# Maternal Transcripts of Hox Genes Are Found in Oocytes of *Platynereis dumerilii* (Annelida, Nereididae)

**DOI:** 10.3390/jdb9030037

**Published:** 2021-09-04

**Authors:** Georgy P. Maslakov, Nikita S. Kulishkin, Alina A. Surkova, Milana A. Kulakova

**Affiliations:** 1Department of Embryology, St. Petersburg State University, Universitetskaya nab., 7-9, 199034 Saint-Petersburg, Russia; St070965@student.spbu.ru (G.P.M.); st069336@student.spbu.ru (N.S.K.); st069328@student.spbu.ru (A.A.S.); 2Laboratory of Evolutionary Morphology, Zoological Institute RAS, Universitetskaya nab., 1, 199034 Saint-Petersburg, Russia

**Keywords:** maternal RNA, Hox genes, Annelida, *Platynereis dumerilii*, oogenesis

## Abstract

Hox genes are some of the best studied developmental control genes. In the overwhelming majority of bilateral animals, these genes are sequentially activated along the main body axis during the establishment of the ground plane, i.e., at the moment of gastrulation. Their activation is necessary for the correct differentiation of cell lines, but at the same time it reduces the level of stemness. That is why the chromatin of Hox loci in the pre-gastrulating embryo is in a bivalent state. It carries both repressive and permissive epigenetic markers at H3 histone residues, leading to transcriptional repression. There is a paradox that maternal RNAs, and in some cases the proteins of the Hox genes, are present in oocytes and preimplantation embryos in mammals. Their functions should be different from the zygotic ones and have not been studied to date. Our object is the errant annelid *Platynereis dumerilii*. This model is convenient for studying new functions and mechanisms of regulation of Hox genes, because it is incomparably simpler than laboratory vertebrates. Using a standard RT-PCR on cDNA template which was obtained by reverse transcription using random primers, we found that maternal transcripts of almost all Hox genes are present in unfertilized oocytes of worm. We assessed the localization of these transcripts using WMISH.

## 1. Introduction

Genome-wide studies that investigate developmental processes have shown that Metazoan genomes are pervasively transcribed in oogenesis to create the complexity necessary for early development, that is, during the period when the zygotic genome is not yet working. Mammalian oocytes contain from 0.3 ng (mouse, human) to 2 ng (cow) of RNA, of which 10–15% are mRNAs [1,2,3]. For comparison, a somatic cell contains only 10 to 30 pg of total RNA, depending on the type.

Maternal factors of *Drosophila* (such as Nanos, Caudal, Bicoid, Hunchback) are at the top of the regulatory cascade that establishes, regionalizes, and patterns the anteroposterior axis of the embryo [4,5]. Pair-rule genes, segment polarity genes and Hox genes are under the direct or indirect control of these factors. Thus, the position of Hox genes within this regulatory hierarchy is several steps below that of maternal genes.

In vertebrates, Hox genes are not expressed in totipotent and pluripotent cells because this expression induces differentiation. In mammalian embryonic stem cells, Hox loci have an ambivalent epigenetic status. Their histone code contains both repressive and permissive tags. Thus, cells do not express the Hox genes, but can quickly begin to do it in the case of additional permissive signals [6]. It is known that among the maternal transcripts of mammals there are mRNAs of Hox genes [7]. Since the Hox loci of the zygote do not function at the earliest stages of development, the early function of these genes, realized by maternal RNA in oocytes and early embryos, should be very different from their later function [7]. In order to investigate these assumptions, we started to analysis the Hox genes of the marine annelid worm *Platynereis dumerilii* (*Pdum*). This animal has 11 Hox genes [8,9], one of which (*Pdum-Post1*) is expressed in chaetal sacs and, apparently, has lost its Hox function [9]. We obtained cDNA from worm oocytes and, using gene-specific primers for ten true Hox genes, revealed by RT-PCR that nine of them have maternal transcripts. We showed the presence of these maternal transcripts for eight genes using the WMISH method. In addition, we have resorted to strand-specific reverse transcriptions (RT) and found sense and antisense transcripts for *Pdum-Hox1*, *Pdum-Hox4*, and *Pdum-Hox7* genes. Apparently, maternal RNAs of Hox genes occur not only in mammals, but rather their occurrence is a widespread phenomenon.

## 2. Materials and Methods

### 2.1. Collection of Material and Fixation of Oocytes for In Situ Hybridization

The material for the study was mature oocytes obtained from adult females of *Platynereis dumerilii*. To stimulate spawning, the animals were placed into a small container with fresh seawater. Then, through a syringe with a filter (d = 0.22 um), we added drops of water from the container, where mature males were contained, already stimulated by engaging with other females. The collected oocytes were checked for intactness (the fertilization membrane should not flake off in unfertilized cells). All material was divided into two parts—for RNA isolation and for in situ hybridization. Oocytes for in situ were fixed with 4% PFA in PBS/0.2% Tween20. After fixation (24 h), they were dehydrated stepwise and stored into 70% ethanol at −20 °C.

### 2.2. Isolation of RNA and Synthesis of cDNA

Total RNA was isolated from *Pdum* oocytes using liquid nitrogen and QIAzol Lysis Reagent (QIAGEN, Venlo, The Netherlands) using the manufacturer’s protocol. The quality and concentration of RNA was checked by electrophoresis in agarose gel and on a spectrophotometer. The samples that were used for RT had A 260/280 ratio 1.95-2. Contaminating genomic DNA was removed from RNA preparations with a DNA-free kit (Ambion, Invitrogen, Waltham, MA, USA). Random primers (10N) and gene-specific primers were used to generate the first strand of cDNA. For random- and gene-specific RT we used two high temperature reverse transcriptases—Maxima (TF Scientific, Waltham, MA, USA) and SuperScript III (Invitrogen, Waltham, MA, USA), respectively. Reverse transcription reactions were performed at 55 °C according to the manufacturer’s protocols. It is worth noting that *P. dumerilii* oocytes contain many polysaccharides required for the formation of the fertilization membrane. These polysaccharides are not removed by reprecipitation or column purification, so despite the good A 260/280 ratio and the good quality of the samples seen on electrophoresis (Appendix A), RNA preparations from oocytes have considerable viscosity.

### 2.3. RT-PCR

The cDNA templates obtained in the previous step were used in the RT-PCR reaction with gene-specific primers for *Pdum*-Hox genes. The same RNA sample used to obtain cDNA was subsequently used as control for the absence of genomic DNA in an amount equivalent to the residual RNA in the cDNA reaction mix. A null control without a matrix was set for each sample to exclude accidental contamination. We used the highly productive DreamTaq DNA Polymerase enzyme (TF Scientific, Waltham, MA, USA) in the PCR reaction. For each gene, the optimal temperature annealing of the primers was selected based on the Tm Calculator (TF Scientific) and empirical experience. For each PCR, 1–2 μL of cDNA were used. We achieved the best results at 34–36 cycles of amplification. These are high values, which can lead to a nonspecific background, but we believe that we are protected from false-positive results by clean controls and size matching of the bands. We think that the efficiency of the PCR reaction can be partially inhibited by the high content of rRNA and tRNA in the cDNA samples from the oocytes (Appendix A). In addition, such cDNA contains the polysaccharides discussed above. The PCR results were evaluated in 1.5% agarose gel and documented by the gel documentation system GDS-8000 System (UVP, Inc., Analytik Jena GmbH, Jena, Germany). A list of primers and the Hox-gene sequences to which they were constructed is presented in Appendix A.

### 2.4. In Situ Hybridization

We used the standard in situ protocol described earlier in the work with *Chaetopterus* sp. [10]. It was adapted for nereid polychaetes with minimal changes. Digoxigenin-labeled RNA probes were prepared according to the manufacturer’s protocol (Roche, Basel, Switzerland). Hybridization was carried out at 65 °C. BM-purple (Roche) was used as a chromogenic substrate to localize the hybridized probe. The results were imaged on DMRXA microscope (Leica Microsystems GmbH, Wetzlar, Germany) with a Leica DC500 digital camera under Nomarski optics. A list of Dig-RNA probes with size and location within the Hox sequences is presented in Appendix A.

## 3. Results

Oocytes of *Pdum* are ellipsoidal and inside these cells, large lipid droplets can be traced, lying along the equator around the smaller of the axes [11]. We collected intact (unfertilized) oocytes of the worm and investigated the presence of maternal Hox gene matrices in them by RT-PCR and in situ. In the first step, we used cDNA, which was obtained using random primers (10N), and we found transcripts of nine Hox genes except *Pdum-Post2* (Figure 1A; Appendix A). We used two sets of primers, some of which are complementary to the 5′ and 3′-exon regions and some to flank the conserved intron located between these exons. Primers to the 5′ and 3′-exons have revealed *Pdum-Hox1*, *Pdum-Hox2*, *Pdum-Hox3*, *Pdum-Hox4*, *Pdum-Hox5*, *Pdum-Lox5*, *Pdum-Hox7*, *Pdum-Lox4,* and *Pdum-Lox2* transcripts (Figure 1A). Intron flanking primers have revealed *Pdum-Hox1*, *Pdum-Hox2*, *Pdum-Hox3*, *Pdum-Hox5*, *Pdum-Lox5*, and *Pdum-Hox7* (Appendix A). We tested primers for *Pdum-Post2* on cDNA from the tails of the juvenile worms *P. dumerilii*, and were convinced of their validity, since this gene is clearly transcribed at this stage (Appendix A). In the next step, we tested how the standard in situ protocol (which is used on larvae and juvenile *Pdum* worms) is appropriate for oocytes. We found that endogenous phosphatases (oocytes are rich in them) are successfully inhibited by the high hybridization temperature (65C) and acidic pH (4.5) of the hybridization buffer (Appendix A). Antibodies to digoxigenin (Anti-Dig) at a concentration of 1: 5000 do not bind to the surface and internal structures of oocytes (Appendix A). We selected a positive control, a worm’s gene homologous to the vertebrates’ gene *Acox3* (Peroxisomal acyl-coenzyme A oxidase 3), involved in the degradation of fatty acids during oocyte maturation in mice [12]. According to our data, the *Pdum-Acox3* is transcribed in *Pdum* oocytes (Appendix A). It is notably that the *Pdum-Acox3* transcripts are localized in the perinuclear cytoplasm with a slight displacement towards one of the poles of the short axis (Appendix A). Antisense probes for Hox genes revealed not so bright, but fundamentally similar transcription without pronounced polarization (Figure 1A). We did not find the *Pdum-Post2* and *Pdum-Hox3* transcripts by in situ. In the case of *Pdum-Hox3*, this contradicts the RT-PCR data. It is possible that the intensity of the *Pdum-Hox3* signal in the oocytes is very low and can only be assessed by RT-PCR. We synthesized sense probes for most Hox genes, and found that they show a weak signal (Figure 1B), which nevertheless turned out to be stronger than the antisense signal in the *Pdum-Acox3* control (Appendix A). We estimated the in situ signals intensity using Fiji software according to the protocol described by Dobrzycki et al., 2020 [13] and used the antibody control (Anti-Dig +) and the antisense signal of *Pdum-Acox3* as background for substraction. According to this estimate, our data differ from the background (Appendix A—Appendix A).

In the next step we synthesized cDNAs using forward (F) and reverse (R) gene-specific primers. Such strand-specific RT allows one to synthesize cDNA strands, which are complimentary to sense and antisense RNAs respectively. According to our data, at least three Hox genes—*Pdum-Hox1*, *Pdum-Hox4*, and *Pdum-Hox7*, are transcribed or stored in oocytes as sense and antisense RNAs (Figure 1C). We cannot state clearly that there are no antisense transcripts of other Hox genes, because chain-specific RT-PCR is not a perfect method, and its sensitivity decreases when using oocyte RNA. In particular, we could not find out from which particular strand the *Pdum-Hox5* transcript is read from in oocytes, although a similar (parallel) experiment on RNA from juvenile worm tails answers this question (Appendix A). The search is also complicated by the fact that splicing of sense and antisense RNAs is carried out at different sites. Probably, Northern blotting would be the best tool for analysis of sense and antisense oocyte transcripts.

## 4. Discussion

*Platynereis dumerilii* is an appropriate model object for evolutionary aspects of early development study. This is a spiralian animal with stereotypical cleavage, which is well traced at the single cells level. These cells have different size and different development potency. This difference is conditioned by segregation of maternal determinants.

It is important that *Pdum* keeps many features which are ancestral for the spiralian animals. In particular, full set (complement) of 11 Hox-genes was found in its genome. This set had been present in last common ancestor of Lophotrochozoa [14,15]. There is a more intriguing feature—the exon-intron organization of *Pdum* genome is similar to that in vertebrates [16]. Moreover, during development of the brain, eyes, and neurosecretory centers, this sea worm uses the same transcription factors repertoire as vertebrates [17,18,19]. It suggests that *Pdum* not only keeps ancestral features of Spiralia, but also is close to ancestor of all Nephrozoa. That is why our object is ideal for study of maternal determinant’s functions in development.

Detailed and comprehensive analysis of early worm’s development transcription dynamics was published in 2016 [20]. In that work seven stages of development were studied, from zygote (2 hpf) to early protrochophore stage of ~330 cells (14 hpf). There were identified 13,160 protein-coding genes with the significant transcription level at least at one of the investigated development stage (FPKM > 1). Using the cluster analysis, authors found that 4302 genes are represented in zygote as maternal matrices. These RNAs degrade with different intensity and are almost completely depleted at 10 h of development. In that study the oocyte stage was not investigated, so we do not know how comparable their transcription landscapes is. However, one maternal transcript, which we found, *Pdum-Hox1*, is also described for zygote [20]. *Pdum-Hox1* isoform, which was annotated in Chou et al. study [20] as a *Hox-B1a*, hit in the cluster with majority of other maternal transcripts. Among the zygote maternal transcripts, small amounts of *Hox2* and *Hox7* mRNA are detected. Other Hox-transcripts (*Hox3*, *Hox4*) are clustered with zygotic genes. It is possible, that main part of maternal RNAs of Hox-genes quickly degrades just after fertilization.

Genes with maternal effect usually are crucial for the early development. In *Drosophila* and vertebrates (*Xenopus, Danio*) maternal determinants set embryos’ spatial coordinates. In sea urchin maternal effects, mRNA and proteins of signal pathways (Wnt and Notch) determine early segregation of cell lines. The Hox-genes’ functions in the early development are enigmatic, because the A-P axis regionalization is a late event. In mammals there were found polyadenylated maternal mRNA of several Hox-genes (HOXD1, HOXA3, HOXD4, HOXB7, HOXB9, and HOXC9) and at least one maternal Hox-protein (HoxB9) [7,21]. The protein HoxB9 has a complex dynamical pattern in extraembryonic tissues and probably is co-opted in pre-gastrulation development program. Other Hox genes might be implicated in the control of oocyte maturation [7]. It is interesting that several genes of different paralogical groups are involved in this process simultaneously.

Maternal RNAs of all cluster Hox-genes are found in oocytes of *Strigamia maritime* (Myriapoda) [22]. They are present as a number of alternative transcripts, some of which do not have an ORF (some abd-A isoforms and all Abd-B isoforms). Their functions have not been studied yet.

Thus, maternal transcripts of Hox-genes were found in mammals, in myriapoda, and in annelid. It may be a coincidence, but it is very strange that Hox-genes are co-opted in the process not individually, but by sets of most paralogs (Figure 2).

It seems that they are necessary here for some crucial evolutionary conservative function. If maternal RNAs are translated in functional Hox-proteins, then their action in oocytes cannot be paralog-specific. It is known that all Hox-proteins of *Drosophila* suppress autophagy by repression of key genes-participants [25]. During the oocytes’ maturation, autophagy can occur in cells, which provide oocytes by nutrient reserves. Oocytes of *Pdum* are formed in body cavities and they have contact with all surrounding tissues. If there is a signal that can induce autophagy, then oocytes must have a protection against it. It seems logical, but in reality, this hypothesis simplifies reality too much, because normally autophagy should occur both in oocyte and in embryo [26]. It is possible that maternal transcripts of Hox-genes are not translated and are necessary for epigenetic adjustment of zygotic genome. In particular, transcriptional silence of Hox-loci in early development of vertebrates requires pretuning. Maternal transcripts (sense and antisense) may be involved in this process.

## Figures and Tables

**Figure 1 jdb-09-00037-f001:**
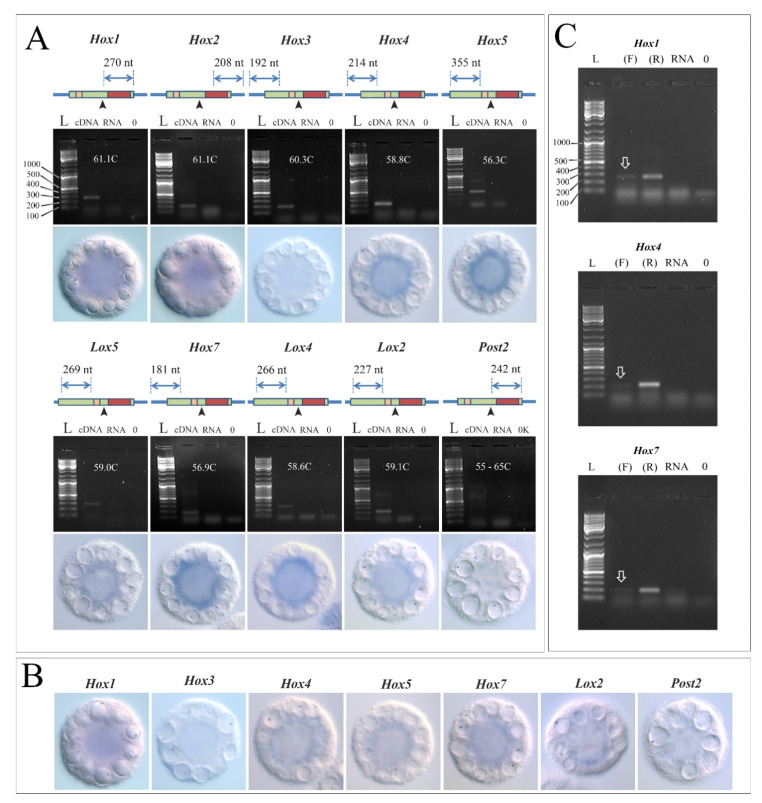
RT-PCR and in situ hybridization results for ten *Pdum* Hox genes. (**A**) Results of RT-PCR and in situ hybridization with antisense Dig-probes to Hox transcripts. Schematic projection of primers onto Hox transcripts. The expected size of the PCR-fragments is indicated above the double-headed arrow. Red rectangle—homeobox; yellow rectangle—hexapeptide sequence; coding region is marked in green, 5′ (left) and 3′ (right) UTRs are marked in blue; the black arrowhead indicates the position of the intron. Scaling between amplified fragments and gene sizes is not respected in the schemes. cDNA from oocytes was obtained using random primers. L-Ladder (GeneRuler DNA Ladder Mix); RNA—genomic DNA control; 0—zero control, where water was used instead of the matrix. The white numbers indicate the annealing temperature. The additional high bands visible at the 800 bp level are not a product of amplification because they are present in the control samples that do not contain Taq-pol (Appendix A). (**B**) Results of in situ hybridization with sense Dig-probes. (**C**) Strand-specific RT-PCR. L—Ladder, (F)—Forward primer; (R)—Reverse primer; 0—zero control. White arrows mark bands synthesized from antisense transcripts.

**Figure 2 jdb-09-00037-f002:**
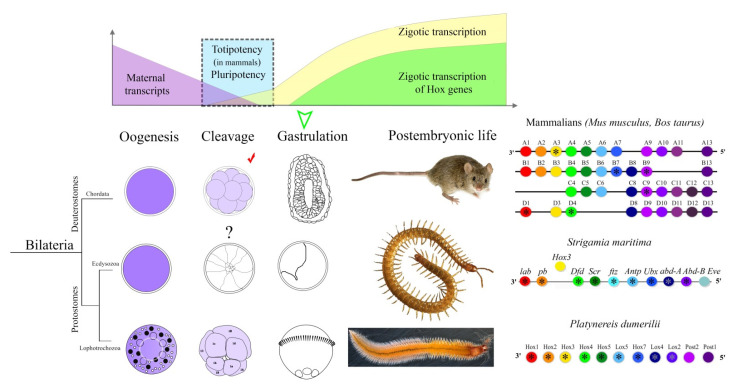
The dynamics of development of bilateral animals in the coordinates of the work of the maternal and zygotic genomes. The upper part of the illustration shows a graph of the activation of the zygotic genome, which is common to many bilaterian [23,24]. Zygotic transcripts of Hox genes (marked in green) are detected after the cleavage stage, at the moment of the onset of gastrulation. Up to this point, there is an epigenetic prohibition on their work, since they violate the state of totipotency and pluripotency. Maternal transcripts of Hox genes (marked in lilac) are present in oocytes animals from all three evolutionary lineages of Bilateria. These RNAs are degraded, but in some cases are found during cleavage (mammals and *Platynereis*) and are even present as proteins (red check mark; HoxB9 in mammals). Hox clusters of animals with Hox-positive oocytes are shown on the right side of the figure. Asterisks mark Hox genes whose transcripts were found in oocytes. Solid line between genes indicates physical linkage where shown. Animal photos copied from Wikipedia.

## Data Availability

Not applicable.

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
