# Peer review of "Maternal Transcripts of Hox Genes Are Found in Oocytes of Platynereis dumerilii (Annelida, Nereididae)"

_jdb, 2021, doi:10.3390/jdb9030037_

Round 1

Reviewer 1 Report

The authors report expression of HOX genes in the annelid Platynereis in oocytes. The data are believable, although not the latest technologies are used, and some unexplained differences remain (see points below). With qPCR or better primer placement the results could be improved, but I am not sure if they authors have the possibility to do this.

In general I am inclined the accept the paper, but the authors should try to address as many of my comments as possible prior to publication.

“errant annelida Platynereis”   I think it should be either “Annelida” in capital letters (Latin), or annelid (the English adjective). Here in this sentence “annelid” would flow better. Check this throughout the manuscript.

“reversion”   is that a standard expression? -> reverse transcription using random primers.

“is realized in “   what do the authors mean here, explain better. More references about early transcription than ref nr 1 (which is rather old)

M&M: “The original RNA” what is that? The RNA before treatment with DNA-free kit? Be more precise.

Can the authors please check the location of the primers. Where are the intron positions in the HOX Genes? They mention that the genes have the usual gene structure, usually there is an intron between the Hexapeptide and the homeobox. If the authors would have placed the primers flanking an intron, any positive RT-PCR result would come from a spliced product, and could not be genomic. It would have been a nice control as well. Did any of their primer pairs span an intron? This would make the results better, even though the authors treated with DNA-free kit.

Since the authors found sense and antisense transcripts for some of the genes, I could be speculated that there is general unspecific transcription (unspliced) happening in Platynereis oocytes. Such general random gene-unspecific transcription has also been observed in other species.

Do the authors have access to a qPCR machine? With this they could perform quantitative PCR. They observed some differences between their in situ hydridiation and PCR results (e.g Hox3). Likewise for the sense and antisense transcripts. With qPCR they could better quantify their results. The authors used Acox3, but this looks like it is transcribed much higher. If qPCR were to be done, also some less expressed genes could be examined, to put expression levels in relation to some other genes.

“part of which hasn’t ORF “ => some of which do not have an ORF.

Author Response

Dear reviewer,

Thank you for your comments. We have made the necessary changes to the text and we hope it has improved.

  1. We have fixed terminological errors:

- changed "annelida" to "annelid"

- removed the slang ("reversion" was replaced by "reverse transcription (RT)")

- «part of which hasn’t ORF » was replaced by «some of which do not have an ORF».

  1. We changed the first paragraph of "Introduction" and inserted references to more modern works.
  2. In M&M we used the unfortunate expression “The original RNA”. We had in mind the same RNA we used for RT, and we make a correction: «The same RNA sample used to obtain cDNA was subsequently used as a control for genomic DNA in an amount equivalent to the residual RNA in the cDNA reaction mix.»
  3. Comments about primer positions and possible nonspecific (unspliced) transcription:

Typically, Hox genes have one intron located between the hexapeptide sequences and the first helix of the homeodomain. We do not know the intron-exon structure of the Platynereis Hox genes because we are using sequences derived from cDNA. We extrapolate to Pdu data from another nereidid (Alitta virens) from which we have genomic sequences. The A. virens Hox genes do have introns in a conservative position. If these worms are similar, then most of our Pdu-primers are located in the regions of either the first or second exons (not through an intron). It happened by accident. Perhaps this is even not bad, because different isoforms of Hox transcripts can differ at the splicing level, and this would introduce confusion into the analysis. Nevertheless, we took your advice and staged an experiment with intron-flanking primers to all of the Hox genes except Hox4 (they were not in our primers collection and we would not have had time to buy them). The data are presented in Supplementary 2.  Apparently, six genes out of nine are represented in the oocytes by RNA templates that are spliced in the expected manner.

Of course, these data still do not guarantee that the expression of Hox genes in oocytes is specific and is needed to perform any functions. A well thought out experiment is needed to answer this question.

  1. Initially, this work involved obtaining qPCR data, but lockdown left us access to only one resource center, which specializes in microscopy and does not have a qPCR-machine.
  2. We used Pdu-Acox3 as a positive control for in situ because it had already been cloned by us earlier for another study and it was just convenient. According to Chou et al., 2017, RNA of Acox3 is at 2hpf, which means it is definitely maternal. The quantitative analysis carried out in this work showed that there are even fewer Acox3 transcripts than Pdu-Hox1, but we do not know anything about the rate of maternal RNA degradation after fertilization and about the difference in this rate between RNAs of different genes.

We attach a table with the primer sequences and additional information about the probes and the Hox sequences themselves (Supplementary 1).

We also made changes to the text and illustrations according to the recommendations of other reviewers.

Thank you for your attention to our work. 

Milana and Lab

Reviewer 2 Report

Maternal transcripts of Hox genes are found in oocytes of Platynereis dumerilii (Annelida, Nereididae)

This manuscript aims to establish the existence of maternal transcripts of Hox genes in the unfertilized oocytes of Platynereis dumerilii. The senior author of the manuscript previously described the expression of these genes during larval development of this model organism (Kulakova et al., Dev Genes Evol. 2007). Here, the authors employ RT-PCR and in situ hybridization experimental approaches to detect Hox transcripts. Although intriguing, the results presented are not sufficient to support the authors claims. Important methodological information is missing, and the specificity of the results is not conclusively established. Finally, the English and overall style of the manuscript, the discussion section in particular, should be thoroughly revised. For the reasons stated, I cannot support the publication of this manuscript.

Specific comments:

  1. The panels in Fig. 1 are too small for proper visualization.
  2. There are important concerns regarding the specificity of the fragments amplified by RT-PCR:
    1. The list of primers used for RT-PCR is missing. Although the authors provide a schematic representation of the amplified fragments, the specific primer sequences are indispensable to replicate the experiment, and thus should be provided.
    2. In some cases, an additional amplified band of higher molecular weight is visible (e.g., Hox5 and Lox2). This is not acknowledged by the authors and could suggest poor primer specificity.
    3. The amplified fragments should have been sequenced for confirmation (not performed).
  3. There are important concerns regarding the specificity of the staining obtained by in situ hybridization:
    1. Description of the in situ hybridization probes used should be provided.
    2. In some cases, the authors detect staining when using sense probes. This is usually an indication that there is lack of specificity in the results obtained with the antisense probes. The authors argue this is not the case and apply RT-PCR to show the presence of antisense RNAs in the oocyte. However, given that the amplified bands obtained with the forward primer (F) in Fig.2 are very faint and that the sensitivity of PCR is significantly higher than that of in situ hybridization, this reviewer is not convinced that the staining obtained in Fig.S3 is not an artifact.

Author Response

Dear reviewer,

We have tried to make changes that are possible to make this article potentially printable.

We increased the size of the main illustration and added data from other primer sets (Supplementary 2).

We have added a list of primers and probes, as well as marked their positions within the Pdum Hox gene sequences (Supplementary 1).

Additional bands that can be seen on some electrophoresis lanes are not a result of PCR as they are present in samples that do not contain Taq-polymerase. This is a feature of RT on oocytic RNA, which is very rich in 28S and especially 18S. When the RT temperature rises, they are destroyed, but this is not always optimal for primers. We have a draft snapshot of control sample that contains all the components of the PCR reaction except for the polymerase.

We will not be able to sequence fragments until autumn (resource center is closed). Using Southern blotting, we transferred two of them (Hox2 F2R2 (345bp) and Lox2 F1R1 (266bp)) to a nylon filter and hybridized with the corresponding probes, including cross-hybridization. Since the bands hybridize specifically and have a predicted size, we believe that the RT-PCR signal is specific.

We are not convinced that PCR is significantly more sensitive than WMISH if used oocytes cDNA template. Unfertilized oocytes contain a large amount of polysaccharides, which are needed to form the fertilization membrane. They are not removed by reprecipitation and not cleaning on columns and RNA has a noticeable viscosity. It is possible that the presence of these polysaccharides reduces the efficiency of the PCR reaction.

Besides, we have reason to think that antisense RNAs may be in oocytes, because we have already encountered them at other stages of Platynereis development and cloned some of them.

 PS. There were illustrations in this letter, but they were not attached, so we are attaching the letter with pictures. 

Thank you for your attention to our work.

Sincerely,

Milana and lab

Reviewer 3 Report

The manuscript of Maslakov et al is addressing in a descriptive manner a very interesting aspect in the Hox field and they use the annelid Platynereis dumerilii as a model system. The striking finding of the paper is that maternal transcripts of Hox genes can be found in the unfertilized oocytes and even more interesting is the fact that sense as well as antisense transcripts are maternally loaded.

For me the findings are very novel and interesting, which may open a total new aspect of Hox function, regulations and requirements to ensure normal development. I have some major and minor comments that may improve the results of the paper.

Major comment:

  1. In the introduction the authors say that the worm has 11 Hox genes and in the results only 10 are tested. What happened to the last one? In addition, in line 96 the authors claim that transcripts of all the Hox genes could be detected and only Post1 is not seen, Figure 1 shows a different picture, where also Hox3 is missing and further on, later in the manuscript the authors say that Hox3 is undetectable. This contradictory has to be addressed.  
  2. To clearly show that Hox genes are expressed and some are not I would recommend to perform a quantification of the in situ expression signal to discriminate significantly between no, little and strong expression. A FIGI bases analysis would be sufficient.
  3. In order to bring key findings together, I would like to encourage the authors that they move the results form Fig S3 into the main figure, since these experiments are key findings. In addition, it would be nice to see the expression of antisense and sense from all Hox genes (present or not).

Minor comments:

  1. There are some unclear sentences in the abstract, which need revision.

Line 11: The work of the Hox … The sentence needs to be better connected to the previous one and the following; like: “Their activation is required for the correct differentiation of cell lines, but in contract to that function stands the potential of the Hox proteins to decrease the differentiation ability of stem cells.”

Line 19-20: part in brackets (which was obtained …) is not needed in the abstract.

Line 22-24: The sentence dilutes the story and makes it sound like if the results are not true and the authors don’t believe in them. It should be removed.

  1. The introduction:

Line 28-30: the first sentence should be reorganized, like: “Genome-wide studies that investigate developmental processes have shown that most of the genetic information …”

Line 41: There is a connection missing between the two sentences. It could be one filling word or an additional sentence.

Line 44: There is a connection missing between the two sentences. “… very different from their later function. In order to investigate these assumptions, we started to analysis the Hox …”

  1. In the results:

Line 114-116: That sentence is not needed and can be deleted. The authors point already to the supplementary figure in the precious sentences.

  1. In the discussion:

Line 172: I would add an “is” to the end. “… their transcription landscape is. …”

Author Response

Dear reviewer,

Thank you for your comments and attention to our work. We have made changes to our article and we hope this has improved it.

  1. In Introduction, we explained the situation with the number of Hox genes and explained why we did not take Pdum-Post1 into the analysis:

«In order to investigate these assumptions, we started to analysis the Hox genes of the marine annelid worm Platynereis dumerilii (Pdum). This animal has 11 Hox genes [6, 7], one of which (Pdum-Post1) is expressed in chaetal sacs and, apparently, has lost its Hox function [7]. We obtained cDNA from worm oocytes and, using gene-specific primers for ten true Hox genes, revealed by RT-PCR that nine of them have maternal transcripts. We showed the presence of these maternal transcripts for eight genes using the WMISH meth-od. In addition, we have resorted to strand-specific reverse transcriptions (RT) and found sense and antisense transcripts for Pdum-Hox1, Pdum-Hox4, and Pdum-Hox7 genes. Apparently, maternal RNAs of Hox genes occur not only in mammals, but this is a wide-spread phenomenon.»

2. We have tried to figure out how signal intensity is measured in Fiji, but this is a very difficult task for us since we have never done it before. In Supplementary3, we inserted tables with the numerical characteristics of the signal and plots based on these numbers. We used the antibody control (Anti-Dig +) and the antisense signal Pdum-Acox3 as the background signal.

3.We inserted data on antisense transcription into the main illustration of the article and enlarged the illustration itself. We hope this has become clearer.

4. The data on sense and antisense transcription of other genes raises many questions that we hope to resolve with other methods. So far, we have draft results for several other genes:

Most likely, several Hox genes also have antisense transcripts. The situation is complicated by the fact that these transcripts can be spliced differently from sense. For example, we do not see antisense transcripts on Hox3 when using primers to the 5'-exon (F1R1; 192 bp), but the pair of primers flanking the intron (F2R2; 447 bp) identify an antisense transcript. It differs in size from the sense and we have to check its specificity. On the other hand, Strand-specific RT-PCR performs worse than Random RT-PCR on templates from oocytic RNA. In particular, we do not see the Hox5 band on F and R matrices, although Random RT-PCR is efficient. We write about this in Supplementary 2.

PS. There were illustrations in this letter, but they didn't attach, so we are attaching the letter with pictures

Best regards,

Milana and Lab

Reviewer 4 Report

In this manuscript Maslakov et al. describe the observation of Hox gene expression in the oocytes of a marine annelid worm Platynereis dumerilii.

As the authors point out in the discussion, this is not the first time Hox gene expression has been observed in oocytes of non-mammalian species. I think the paper would benefit from more explicitly comparing and contrasting the known order of Hox gene expression across Metazoans, rather than separately describing them as they have done. In addition some comparison of the function between species and relation to the expression timing would be beneficial. Both these additions would aid in highlighting to the novelty and relevance of the work presented. The inclusion of a tree diagram annotated with oocyte Hox gene expression status would be an appropriate way to graphical represent the reason why this work would be of interest more generally to a developmental biology audience.

In relation to the design of the study, my confidence in the specificity of the amplification observed in RT-PCR experiments would be improved if the authors showed negative and positive control conditions, using RNA extracted from stages where these genes are known to be expressed (or not) in Pdum. For example, the developmental stages highlighted in the discussion where Hox gene expression has been detected by RNA-seq could be used to inform this. This not only support that the amplification observed isn't the result of inefficient amplification of a closely gene section, but also give some indication of the relative abundance of these transcripts in the oocyte (the bands presented are not bright). A similar argument can also be made to ensure specificity of probe for the in situ hybridisation experiments.

I could not find reference to the number of amplification cycles used for the PCR, which is of course essential information when considering both abundance and specificity of amplification, and must be included. As well as primer and probe sequences.

Author Response

Dear reviewer,

We are grateful for your attention to our work. We have made changes in accordance with your advice and comments and hope that the article has improved.

  1. A summary illustration has been inserted into the Discussion section (Fig. 2)
  2. The primers we used in our work were pre-tested on genomic DNA several years ago. Because when we see a band in a cDNA sample, it matches the expected size, we did not recheck these primers. We decided that with a clean control, they were sufficiently correct. Because the primers for Post2 did not work on cDNA from oocytes, we tested their performance on cDNA from juvenile worm tails and were convinced that they worked (Supplementary 2, Fig. 3)

In the time allowed by the publisher to respond, we used another set of primers that flank the conserved intron between the hexapeptide and the first helix of the homeodomain (Supplementary 2, Fig. 2). Six of the nine genes appeared to be represented by spliced variants. We did not detect Lox4, Lox2, and Post2 transcripts using a new set of primers, which may speak in favor of their absence (Post2) or alternative splicing in the oocytes.

We indicated in the article the number of amplification cycles for the PCR:

«We achieved the best results at 34-36 cycles of amplification. These are high values, which can lead to a nonspecific background, but we believe that we are protected from false-positive results by clean controls and size matching of the bands. We think that the efficiency of the PCR reaction can be partially inhibited by the high content of rRNA and tRNA in the cDNA samples from the oocytes. In addition, such cDNA contains the polysaccharides discussed above.»

It is worth noting that P. dumerilii oocytes contain very many polysaccharides required for the formation of the fertilization membrane. These polysaccharides are not removed by reprecipitation or column purification, so despite the good A 260/280 ratio and the good quality of the samples seen on electrophoresis (Supp.2. Fig.1), RNA preparations from oocytes have considerable viscosity.

We also inserted tables with primer and probe sequences into Supplementary 1.

Best regards,

Milana and Lab

Round 2

Reviewer 1 Report

The authors have undertaken substantial revisions. While it is a pity that no qPCR could be performed, I understand the difficulties in the current situation, and suggest that the paper can published as is. The caveats are explained well enough. 

Author Response

Dear Reviewer,

Thank you very much for your attention to our work. 
We have improved the article through our joint efforts. 

Best regards,
Milana and Lab

Reviewer 2 Report

Maternal transcripts of Hox genes are found in oocytes of Platynereis dumerilii (Annelida, Nereididae)

In response to reviewers’ comments, the authors provide more experimental data and made significant changes to the manuscript. Namely, key methodological information is now provided, and new control experiments are shown. The Figures in the manuscript have also been significantly altered, including the merging of the two original figures and the addition of a new image.

Although this reviewer acknowledges the efforts taken, and that some of the concerns raised have been met at least to some extent, the manuscript remains confusing, and the claims are not fully supported by the data. Much of the current manuscript consists of optimization strategies, which should continue to take place until the data are strong enough to support the claims. As such, I cannot support the publication of this manuscript.

Specific comments:

  1. The authors claim that:

“We did not find the Pdum-Post2 and Pdum-Hox3 transcripts by in situ. In the case of Pdum-Hox3, this contradicts the RT-PCR data. It is possible that the isoform that we see using RT-PCR is not detected by the Dig-probe, which is complementary to the region located closer to the 3’-end of the gene.”

The information provided regarding the probe sequence and primer pairs contradict this statement. In fact, the Hox3 Intron-flanking primer pair encompasses more than half of the probe sequence. The results are consistent with a higher sensitivity of RT-PCR than in situ hybridization (which the authors state is not the case in their hands).

  1. The quality of the isolated RNA and its impact on RT-PCR results remains an unresolved issue (possibly solvable using phenol/chloroform extraction on the RNA and/or the cDNA?)
  2. Controls without Taq-pol are mentioned but not shown in the manuscript. A possible solution to overcome the presence of unspecific bands in the amplification could be to use much less cDNA in the PCR reactions (the amount used is not mentioned).
  3. The schematic representation of the amplified regions above each gene in Fig.1A is misleading. In fact, the drawings are not to scale and, more importantly, they do not correctly represent the amplified region. For ex, Hox3 amplification is represented at the 5’ region of the gene, while according to the sequence information provided (List S1), it is located exactly midway in the gene.
  4. 1A. legend should read “… with antisense Dig-probes…”
  5. Despite the quantification of in situ hybridization signals and their analysis presented in Supp materials 3, #2, these data are not sufficient to ensure specificity of the in situ results. In particular, the authors do not disclaim what level of intensity after subtraction was the cutoff to decide on. They agree that antisense Post2 is not present (with a signal intensity of 10,505 and 11,1, depending on the background subtracted), and claim antisense Lox2 is expressed (signal 13,872 and 14,47). The criterion used is not specified, especially considering that the values obtained for the positive control, Acox3s, are 10 times higher - 138,048 and 138,646, respectively.
  6. The positive control for in situ hybridization was Acox3s, hence Graf S1 and Graf S2 should contain the results for this gene. It would show that the results herein reported are to be taken with reserve since the signal obtained for Hox genes is very low, indicating the need for subsequent validation with other techniques (not convincingly provided in the current manuscript).
  7. The authors acknowledge the frailty of their results regarding the presence of antisense transcripts and mention that “Northern blotting would be the best tool for analysis of sense and antisense oocyte transcripts.” This should be performed if the claim is to be trustworthy.
  8. S4 is not mentioned in the manuscript text.
  9. The organization of the new supp materials is messy and hinders result interpretation.
  10. The new Figure 2 does not represent/summarize the findings of this work, and is more appropriate for a review/comment paper.

Author Response

Dear Reviewer,

We tried to answer most of your questions.

Specific comments:

  1. The authors claim that:

“We did not find the Pdum-Post2 and Pdum-Hox3 transcripts by in situ. In the case of Pdum-Hox3, this contradicts the RT-PCR data. It is possible that the isoform that we see using RT-PCR is not detected by the Dig-probe, which is complementary to the region located closer to the 3’-end of the gene.”

The information provided regarding the probe sequence and primer pairs contradict this statement. In fact, the Hox3 Intron-flanking primer pair encompasses more than half of the probe sequence. The results are consistent with a higher sensitivity of RT-PCR than in situ hybridization (which the authors state is not the case in their hands).

Yes, you are correct. The intron-flanking primers overlap the probe region. We tested this probe at other stages of development and were convinced that it works. Probably the intensity of the in situ signal is lower in oocytes and can only be evaluated by RT-PCR. We made a correction to the text.

2. The quality of the isolated RNA and its impact on RT-PCR results remains an unresolved issue (possibly solvable using phenol/chloroform extraction on the RNA and/or the cDNA?)

Our RNA is of good quality. This can be seen from its spectrophotometric indices and electrophoresis data. The problem is that oocyte RNAs are highly enriched in ribosomal RNAs. The problem could be solved by using oligo-dT primers for reversion, but our matrices are most likely depolyadenylated (like most oocyte mRNAs). We tried using oligo-dT-RT, but the results were worse than when using Random primers. We stopped optimizing RT because we see bands of the expected size and clean controls. 

  1. Controls without Taq-pol are mentioned but not shown in the manuscript. A possible solution to overcome the presence of unspecific bands in the amplification could be to use much less cDNA in the PCR reactions (the amount used is not mentioned).

We made an additional control for Taq-Pol and placed it in the Supplementary 2: Fig.S2.

For each PCR, 1-2 μl of cDNA were used.

  1. The schematic representation of the amplified regions above each gene in Fig.1A is misleading. In fact, the drawings are not to scale and, more importantly, they do not correctly represent the amplified region. For ex, Hox3 amplification is represented at the 5’ region of the gene, while according to the sequence information provided (List S1), it is located exactly midway in the gene.

The schemes we show in Fig.1A are not scaled, because  the gene sequences are much larger than the amplifiable region. We should have written this in the caption of the figure, and we do so. Only the relative position of the primers within the protein-coding region, UTRs, and key domains of the protein are  important here. We have more carefully aligned the boundaries of the amplicons relative to them.  The scheme for Hox3 is correct because F1 falls within the 5'UTR and R1 lies within the 5'-coding region of the transcript. The Hox3 sequence that we used for cloning and primer construction contains a long 5'UTR site and is entirely represented in List S1. It would probably be wrong to cut it off and still refer to its number in GeneBank. To avoid misunderstandings, we decided to highlight the protein-coding regions of all the presented sequences in bold.

  1. legend should read “… with antisense Dig-probes…”

It has been corrected.

  1. Despite the quantification of in situ hybridization signals and their analysis presented in Supp materials 3, #2, these data are not sufficient to ensure specificity of the in situ results. In particular, the authors do not disclaim what level of intensity after subtraction was the cutoff to decide on. They agree that antisense Post2 is not present (with a signal intensity of 10,505 and 11,1, depending on the background subtracted), and claim antisense Lox2 is expressed (signal 13,872 and 14,47). The criterion used is not specified, especially considering that the values obtained for the positive control, Acox3s, are 10 times higher - 138,048 and 138,646, respectively.

This is the first time we have evaluated the intensity of hybridization signals quantitatively. This was a requirement of one of the reviewers and he seems to be satisfied with the result.  Assuming that the antibody wash control and the control with the antisense probe for Acox3 characterize a "no signal" situation, the variants with negative signal values for Hox3as and positive signal values for Post2 may characterize noise. The graphs show that the values for the compared genes are higher than the noise values. Just in case, we took another gene (Pdum-Cad) for which we do not see transcription by RT-PCR and in situ methods. In numerical terms, it is very similar to Post2. Is it worth inserting this data into the Supplementary?

PS: It may be that Post2 and Cad transcripts are present in the oocytes, but both of our methods are insufficient to detect them. In this case, their values are not noisy.

  1. The positive control for in situhybridization was Acox3s, hence Graf S1 and Graf S2 should contain the results for this gene. It would show that the results herein reported are to be taken with reserve since the signal obtained for Hox genes is very low, indicating the need for subsequent validation with other techniques (not convincingly provided in the current manuscript).

The Acox3 gene helped us to put the in situ method on oocytes, but its sense transcript is not suitable for quantitative analysis of Hox genes transcription, because it is not correct to compare the transcription of enzyme and regulatory factor genes. Their physiologically relevant concentrations will be very different. For example, the transcription level of Acox3  is much higher than of HoxA2 in humans (there is a picture in the attached file).

So far, we do not have a suitable positive control (a transcription factor about which we know for sure that its templates are in the oocyte), except for the Hox genes themselves.

  1. The authors acknowledge the frailty of their results regarding the presence of antisense transcripts and mention that “Northern blotting would be the best tool for analysis of sense and antisense oocyte transcripts.”This should be performed if the claim is to be trustworthy.

We could not fully rely on the Strand-specific RT-PCR method, not because it produces false-positive results, but because it is not sensitive enough for total oocyte RNA. That is, in most cases we don't see antisense transcripts, although this doesn't rule them out.  Northern blotting could really help in this situation. We expect to use it in the next step, when we will compare Hox genes isoforms at different stages of development. We think this is a separate big job that will require label selection (P32 or Dig), as well as selection of membrane types and hybridization buffers. Not all of these are available right now.

We assume that antisense transcripts of some Hox are present in oocytes on par with sense transcripts for several reasons:  we see bands at F- Strand-specific RT-PCR for several genes, we see the signal on WMISH, and we know that antisense Hox transcripts are present at other stages of Pdum development. Also, it is quite common for Hox clusters in general to produce antisense transcripts. It does not seem to me that our assumption of antiHox is groundless.

9. S4 is not mentioned in the manuscript text.

We missed the letter S in the picture reference: (Supplementary 2: Fig.4). Correct this to (Supplementary 2: Fig.S5).

  1. The organization of the new supp materials is messy and hinders result interpretation.

We tried to organize additional materials thematically: sequences, RT-PCR data, and in situ data.

  1. The new Figure 2 does not represent/summarize the findings of this work, and is more appropriate for a review/comment paper.

We put Fig.2 in Discussion on the recommendation of one of the reviewers. He has already accepted it as a desired correction.

Best regards,

Milana and Lab

Reviewer 4 Report

The additions and clarifications the authors have made to the manuscript have been useful, and I particularly welcome the discussion figure on Hox gene expression among different lineages. The inclusion of data on flanking primers also adds to the confidence that it is indeed the transcripts of interest that are being detected, and reveals some information about the splice variants present in the oocytes at this stage. I'm happy to accept the manuscript in it's current form.

Author Response

Dear Reviewer,

Thank you very much for your attention to our work.  Your advice and comments were helpful. Our data have gained more weight after additional experiments and the final scheme improves the perception of the article. 

Best regards,

Milana and Lab
